# Time Spent on Mobile Apps Matters: A Latent Class Analysis of Patterns of Smartphone Use among Adolescents

**DOI:** 10.3390/ijerph20156439

**Published:** 2023-07-25

**Authors:** Lucia Fortunato, Gianluca Lo Coco, Arianna Teti, Rubinia Celeste Bonfanti, Laura Salerno

**Affiliations:** Department of Psychology, Educational Science and Human Movement, University of Palermo, Viale delle Scienze, Edificio 15, 90128 Palermo, Italy; lucia.fortunato@unipa.it (L.F.); gianluca.lococo@unipa.it (G.L.C.); arianna.teti@unipa.it (A.T.); rubiniaceleste.bonfanti@unipa.it (R.C.B.)

**Keywords:** problematic smartphone use, adolescent, social media use, screen time, Instagram, TikTok, latent class analysis

## Abstract

The aims of the present study are: (1) to determine classes of adolescents with homogeneous patterns of smartphone or social media use; and (2) to examine the level of distress across the empirically derived profiles. Three hundred and forty adolescents (M_age_ = 15.61, SD = 1.19; 38.2% females) participated in a cross-sectional survey. Participants provided objective trace data on time spent on smartphones and applications, as well as self-reported social media addiction, social media use intensity, online social comparison, emotion dysregulation, and psychological distress. Latent class analysis (LCA) with total smartphone use categorized participants into three classes. Participants in Class 3 (19%) showed a more impaired functioning profile, with a tendency towards social media addiction and greater levels of distress. LCAs with the amount of time devoted to specific applications are more heterogeneous, and results showed that heavy use of social media apps was not consistently connected to the most impaired psychosocial profiles. Although the amount of mobile screen time can be a characteristic of problematic users, the link between social media usage and an adolescent’s psychological characteristics is mixed. More research is needed to explore the interplay between mobile screen time and social media usage among adolescents.

## 1. Introduction

Part of teenagers’ lives takes place online via smartphone. Nowadays, 95% of 13–17-year-olds in the United States use a smartphone and 45% of them report an almost constant connection to the online world [1,2]. The number of smartphone users is highest in China, India and the United States. YouTube is the most commonly used social media platform (used by 95% of teenagers), followed by TikTok (used by 67% of teenagers), Instagram and Snapchat (used by about six out of ten adolescents), and, to a lesser extent, other social media (e.g., Facebook, Twitter, Twitch, WhatsApp) [1]. Growth in the smartphone market has also taken place in European countries. In Italy, more than 95% of adolescents own a smartphone and access the Internet on a daily basis [2]. One of the trendiest, TikTok, went from being used by 28.7% of the sample in one year (2020 survey) to 65% (2021 survey). Although Facebook continues its inexorable descent among ‘teens’ (fewer than 15% now use it), Instagram remains stable (90%). As for WhatsApp, almost universal use is confirmed (over 98%) [3]. With the widespread use of smartphones, there has been a call for examining psychosocial correlates associated with smartphone use, as well as its effect on individual distress. To date, scholars have been mainly concerned with investigating the effects of problematic smartphone use, focusing on its potential addictive behaviors [4,5]. Smartphone use can be problematic because its excessive use may negatively impact several areas of a person’s life [6] and may lead to various mental health problems, especially among adolescents. A growing body of reviews [7,8,9] has revealed that problematic smartphone use is consistently and significantly associated with depression, decreased sleep quality, anxiety and cognitive emotion regulation [10,11]. Problematic smartphone use is often referred to as smartphone addiction [12], which may lead to uncontrolled cravings, withdrawal symptoms and anxiety. However, scholars suggested that problematic smartphone use should not necessarily be recognized as a proper behavioral addiction, given that it may derive from other sources [4]. For example, Billieux et al. [6] suggested that problematic smartphone use is a multi-faceted behavior which is associated with different developmental pathways, such as over-reassurance (e.g., the need for interpersonal contact and reassurance from others), impulsivity (lack of control that triggers dysregulated use) and extraversion (e.g., high sensitivity to reward). Furthermore, it was suggested that smartphone use per se may not be associated with poor mental health, but particular patterns of smartphone-related behaviors and activities (e.g., constantly checking for notifications, watching videos, online gaming, sharing photos, chatting or excessive usage) [13].

Smartphone users in the adolescent age group can be more at risk of problematic smartphone use [4]. It is worth noting that adolescents generally use their smartphones to access social networking services or instant messaging apps [14]. There is recent evidence that engagement with social media can account for the main amount of time spent on smartphones [15]. Furthermore, some overlaps between problematic smartphone use and problematic social media use have been highlighted [13] within the addiction-like model. Regarding messaging apps, WhatsApp has become extremely popular among adolescents, given that it allows the user to keep in touch with others via instant text, voice messages and other multimedia. Recent research has shown that problematic smartphone use and problematic WhatsApp use were strongly intertwined and tended to form a cluster of their own [16]. However, this study of differences between problematic smartphone use and problematic use of specific apps available on smartphones is still in its initial phases [13,17].

In the field of research related to adolescent smartphone use and distress, the effects of problematic smartphone use on mental health are likely to be controversial, bidirectional and related to personal and contextual factors, but also often related to the different methodologies used by researchers [18].

Thus, conclusions regarding the relationships between problematic smartphone use, social media use and adolescents’ mental health remain elusive [19] and more research is needed to examine the interplay between characteristics of adolescent smartphone use and psychological distress. In the current study, we will focus on individual characteristics of functioning (e.g., emotion dysregulation, online social comparison), which can interfere within the relationship between problematic smartphone use and psychological health among young people. Specifically, we will examine different empirically derived adolescent profiles based on their smartphone use and psychosocial characteristics, in order to determine what subgroups may be at greater risk of psychological distress.

### 1.1. Devices and Media Use: Screen Time

Given that excessive smartphone use is an indicator of problematic use, prior research with adolescents has extensively focused on the amount of time spent on smartphones and social media (i.e., mobile screen time). There is limited evidence for negative associations between digital screen engagement and adolescent well-being [20]. A variety of devices (e.g., computer, TV, tablet, smartphone) and uses (e.g., social communication, gaming) can comprise the construct of “screen time”, and there is an ongoing debate about its usefulness in studying adolescents’ mental health [21,22]. Recent longitudinal studies that disentangled the within-person and between-person effects on the relationship between time spent by adolescents on mobile screens and mental health showed no causal connection between these two variables [23,24]. Although mobile screen time is often used as an indicator of problematic smartphone use, recent findings also highlighted how problematic smartphone use (and also problematic social media use) could not be solely defined by mobile screen time (i.e., “how much”), but rather by the type of usage (“how”) as well as by the types of activities that individuals engaged in when using smartphones [25].

It has been hypothesized that elevated mobile screen time and, consequently, extensive social media use, may not necessarily be detrimental to adolescents’ mental health, as frequent social media use may not interfere with life domains relevant to their mental health (e.g., offline socializing with friends or family) [26].

Moreover, intense social media use might represent a normative behavior of adolescents rather than a specific behavior of adolescents with poor mental health. For adolescents, the most common way of measuring the amount of daily connection time is through self-reporting hours. However, self-reporting the amount of time may be subject to recall bias [27,28], and scholars have argued that data from self-reporting problematic smartphone use may make it difficult to examine what type of smartphone activity can be described as problematic [29]. Some concerns have been raised about the validity of self-report measurements of the amount of hours spent online, given the evidence pointing to adolescents’ tendency to under- or overestimate smartphone use [30,31]. Some recent research has moved towards an investigation of the frequency and duration of smartphone use by using objective trace data, i.e., data which can be collected directly through smartphone devices, in order to monitor participant behaviors in real time [32,33].

In summation, screen time appears to foster negligible negative effects on the mental health of adolescents [21,22,23,24,25,26,27,28,29,30,31,32,33,34], and there is a call to examine the motivations underlying smartphone and social media use, as well as their interplay with the individual’s psychological characteristics. The current study aims to fill this gap by providing some evidence of the association between adolescents’ psychological characteristics and their time spent on the most popular mobile applications, such as Instagram, TikTok, and WhatsApp. In the present study, we will examine objective trace data on both (a) overall daily time spent on the smartphone for any activity, and (b) daily time spent on Instagram, TikTok, and WhatsApp. Most of the previous reviews showed that the majority of studies in the field provided the total amount of screen time regardless of the applications used [22]; the link between time spent on each specific application and the adolescent’s characteristics still remains unclear.

### 1.2. Emotion Regulation and Social Media Use

Prior research on problematic smartphone use highlighted its link with dysfunctional emotion regulation strategies [35,36,37]. The theory of compensatory internet use [38] posits that people may increase internet (and smartphone) use to regulate their own negative emotions. Social anxiety and smartphone addiction can be influenced by emotional dysregulation [39]. Emotional regulation skills are, in fact, the basis of good social functioning [40,41]. However, emotion regulation difficulties may be associated with social anxiety [42]. Moreover, both social anxiety [43] and specific emotion regulation strategies such as rumination [36] have been shown to play a prominent role in fostering problematic smartphone use. An association between emotional dysregulation and problematic smartphone use in adolescents and young adults is supported by recent meta-analytic studies [44,45]. However, although a growing body of evidence suggests that emotion dysregulation may lead to both problematic smartphone use [36] and problematic social media use [46], its interplay with patterns of smartphone use among adolescents is yet to be established. Although some empirical evidence suggests that emotion dysregulation may be associated with problematic Facebook use [47,48,49], there is still a dearth of research on the association between emotion dysregulation and time spent on specific applications such as Instagram, TikTok and WhatsApp among adolescents.

### 1.3. Online Social Comparison

Nowadays, adolescents use their smartphones to fulfill their need for social connection through social networking sites (e.g., Facebook, TikTok, Snapchat and Instagram). Social comparison takes place when adolescents assess their own personal value based on comparison with others, i.e., their perception of their position relative to others in different areas of life [50]. Festinger’s social comparison theory [51] argues that a low level of self-confidence is equivalent to a higher rate of social comparison. This particular aspect becomes an important key to understanding the phenomenon in adolescence: in fact, adolescents, according to Erikson [52], are at an important point in their developmental journey; they go through an “identity crisis” and may use peers to obtain social information about themselves in order to construct their identity [53,54]. It was suggested that, in the online context, upward social comparison (i.e., with a superior comparison target) is likely to occur, as on social media, people may be more frequently confronted with the successes than the failures of their online friends [55]. Although there is evidence that online social comparison can result in a decrease in an individual’s well-being [56], most prior studies focused on Facebook and comprised young adults. However, social comparison in social media can be especially important for adolescents, given their widespread comparing of appearance (i.e., how my body looks compared to other people’s) on appearance-focused social media platforms such as Instagram or TikTok [57].

### 1.4. The Present Study

The goal of the present study is to identify distinct groups of adolescents based on their patterns of mobile screen use through Latent Class Analysis (LCA). Specifically, in order to understand how smartphone and social media use is associated with adolescent phenomena such as psychological distress, we have built upon previous research which used clustering techniques to detect specific profiles of smartphone or social media users [15,58,59], to explore whether distinct and meaningful user profiles can be identified. The aims of the present study are: (1) To determine classes of adolescents with homogeneous patterns of smartphone use (i.e., daily overall time spent and time spent on smartphone applications), social media use (i.e., intensity of social media use and problematic social media use), and psychosocial variables (i.e., emotion regulation and online social comparison). For the purposes of this study, we examined the most widely used social networking services or instant messaging apps among Italian adolescents, such as Instagram, TikTok and WhatsApp [14]. (2) To explore whether the adolescent empirically derived profiles are associated with risky psychological outcomes.

Regarding the first goal, we pose the following research questions:What psychological clusters can be identified among adolescents based on their smartphone and social media activities?Do the participants’ psychological characteristics vary across the clusters, according to the types of smartphone application?

Regarding the second aim, we pose the following research question:
Do adolescents with more problematic patterns of smartphone use report a higher level of psychological distress than adolescents with less problematic smartphone use?

## 2. Materials and Methods

### 2.1. Participants and Procedures

Three hundred and forty adolescents (age range = 13–19 years; M = 15.61, SD = 1.19) were included in the study. Participants were recruited from two secondary schools (from the first to the fourth year of high school) in southern Italy (i.e., Palermo and Naples) from March to April 2022. Inclusion criteria were: (a) owning a smartphone, (b) parents having signed the study informed consent document, and (c) Italian language skills sufficient to understand the study questionnaires.

Participants’ sociodemographic characteristics are shown in Table 1. The teachers as well as the parents of the participants were informed about the aims of the study and received consent forms. Adolescents participated in the study voluntarily and data were collected anonymously (i.e., data were anonymized with a single research identification number for each participant). Participants received no compensation and they could leave the study at any point. Using an online survey, questionnaires were completed in the adolescents’ school classrooms, under the supervision of two research assistants. The research adhered to the Italian Psychological Association’s requirements (AIP—ethical standards), and the Declaration of Helsinki.

### 2.2. Measures

The first part of the questionnaire was used to collect information about participants’ demographic characteristics (i.e., age, gender, engagement status, city of residence, and school class). In the next part, questions about participants’ smartphone and social media use were inserted (i.e., time spent on a smartphone and each social media, social media usage, social media addiction, social media use intensity, and online social comparison). Finally, data about difficulties in emotion regulation and psychological distress were collected.

#### 2.2.1. Time Spent on a Smartphone and Social Media

Adolescents were asked to report the time (hours in a day) spent both on all mobile screen time activity (i.e., overall time spent on the smartphone) and on each social media (i.e., Instagram, Facebook, TikTok, Snapchat, and Twitter) and WhatsApp applications. Detailed indications were provided to the participants to objectively trace this information from their IOS or ANDROID devices.

#### 2.2.2. Social Media Addiction

The Bergen Social Media Addiction Scale (BSMAS) [60,61] is a six-item self-administered scale of an individual’s level of addiction-like use of social media (e.g., “How often during the last year have you used social media in order to forget about personal problems?”). Each item is rated on a Likert scale from 1 (Very rarely) to 5 (Very often), and higher scores indicate higher addiction-like use of social media. The BSMAS demonstrated a high level of internal consistency (α = 0.748).

#### 2.2.3. Social Media Use Intensity

Respondents’ social media activities were assessed based on four items [24,62] (e.g., “How many times per day do you send a message, photo or video via your smartphone, for example, a WhatsApp, Chat, Snapchat, or SMS?”). Items were rated on a 7-point Likert scale, from 1 (Never or less than once and Less than once for the first three items and the fourth item, respectively) to 7 (More than 40 times and More than 80 times for the first three items and the fourth item, respectively). In the present study, the ratings of the four items demonstrated high levels of internal consistency (α = 0.809) and were totaled in order to calculate an aggregate score (higher scores indicate higher levels of social media use intensity).

#### 2.2.4. Online Social Comparison

The IOWA-Netherlands Comparison Orientation Measure (INCOM) [63] is a brief 11-item measure of an individual’s tendency to evaluate him/herself by comparing with others (e.g., “I often compare myself with respect to what I have accomplished in life”). Participants responded on a Likert-type scale with 5 response options, which ranged from 1 (Strongly Disagree) to 5 (Strongly Agree). In the present study, the scale showed good internal reliability (Cronbach’s α = 0.766).

#### 2.2.5. Difficulties with Emotion Regulation

The Difficulties in Emotion Regulation Scale (DERS-Short Form) [64,65] is an 18-item self-report measure of emotion dysregulation (e.g., “When I am upset, I become embarrassed for feeling that way”). Responses were formulated on a Likert-type scale with 5 response options which ranged from 1 (Almost never) to 5 (Almost Always). In the present study, the DERS-SF showed good internal reliability (Cronbach’s α = 0.868).

#### 2.2.6. Psychological Distress

The Young Person’s CORE (YP-CORE) [66] is a brief 10-item measure of psychological distress in young people (11–16 years) (e.g., “My thoughts and feelings distressed me”). Items were rated using a five-point Likert scale, ranging from 1 (Not at all) to 5 (Most or all of the time). In the present study, the YP-CORE showed good internal consistency (Cronbach’s α = 0.740).

### 2.3. Plan of Data Analyses

Preliminary analyses (i.e., Cronbach’s alpha, descriptive statistics, and missing data analysis) were conducted using SPSS software (SPSS 22). All variables had a normal distribution (|Sk| < 2 and |Ku| < 1). There were no missing data on all the variables in the study.

The first LCA was conducted to classify the participants into homogeneous groups of smartphone and social media use patterns (i.e., duration of smartphone usage, social media addiction, social media use intensity, an individual’s tendency to evaluate him/herself through comparison with others, and emotion dysregulation). LCA allocates participants to latent empirically derived groups with homogeneous patterns of observed scores. Three other LCA were conducted, replacing the variable “time spent on the smartphone” with the time spent on the most-used social media platform (i.e., TikTok and Instagram; cf. Table 1) and in a messaging service (i.e., WhatsApp). We tested models with one to five latent classes through Mplus software (Mplus 7) using maximum likelihood estimation (which produces values that approximate the observed values [67]). In order to determine the number of classes, multiple indexes were used (i.e., the Bayesian information criterion, the sample size adjusted BIC, the entropy, the Lo–Mendell–Rubin likelihood ratio test, the bootstrap likelihood ratio test, the smallest class size, and the clinical meaning of the empirically derived homogeneous groups). The most meaningful and parsimonious model had the following indices: lower BIC and ssaBIC; larger entropy (more specifically, entropy may range from 0 to 1 and values ≥0.80 reflect higher classification accuracy [68]), smallest class size >5%; and significant LMR and BLRT [69].

Finally, one-way ANOVAs followed by Bonferroni post hoc tests were conducted to compare the empirically derived classes (obtained through the LCAs) on the level of distress (CORE-YP).

## 3. Results

### 3.1. Results of LCA with Overall Time Spent on Smartphone

Three classes of smartphone users were identified (the LMR LRT for the four-class model was not significant, and the classes identified were clinically distinct, as shown in Table 2).

Class 1 (*n* = 84; 25%) achieved the lowest levels for all the observed variables (except for DERS). Class 3 (*n* = 64, 19%) had the highest levels for all the observed variables. Class 2 (*n* = 192, 56%) had social media use intensity scores similar to Class 3, but in the other indicators, it showed similar scores to Class 1. Descriptions for the three classes are reported in Table 3.

Class 3 showed higher distress levels (CORE-YP) than Classes 1 and 2 (F(3, 337) = 17.559, *p* < 0.001; Class 3 > Classes 1 and 2).

### 3.2. Results with Time Spent on TikTok

Three classes of TikTok users were identified (taking into account the smallest class size in the model with four classes and the clinical meaning of the empirically derived groups; Table 4).

Class 1 (*n* = 68; 20%) achieved the lowest levels for all the observed variables. Class 2 (*n* = 204, 60%) spent a similar amount of time on TikTok to Class 1 but had high scores for all other indicators. Class 3 (*n* = 68, 20%) had higher scores for time spent on TikTok than the other two classes, and DERS, BSMAS and INCOM scores in line with those of Class 2 (Table 5).

A statistically significant difference in distress scores (CORE-YP) was found between Class 2 and Class 1 for CORE-YP (F(2, 337) = 5.548, *p* < 0.01; Class 2 > Class 1).

### 3.3. Results with Time Spent on Instagram

Four classes of Instagram users were identified (taking into account the smallest class size in the model with five classes and the clinical meaning of the empirically derived classes, as shown in Table 4).

Class 1 (*n* = 76; 22%) had the lowest scores for all the indicators (except for DERS). Class 2 (*n* = 44, 13%) spent an average of 2.77 h per day on Instagram and had higher scores than other classes on all other indicators. The third class (*n* = 53, 16%) achieved the highest amounts of time spent on Instagram (8 h) but mild scores for all other indicators. Class 4 (*n* = 167, 49%) spent an average of 1.98 h on Instagram but had mild scores for all other indicators (Table 6).

A statistically significant difference in distress scores (CORE-YP) was found between classes for CORE-YP (F(3, 336) = 9.013, *p* < 0.001; Class 2 > Classes 1, 3 and 4).

### 3.4. Results with Time Spent on WhatsApp

Four classes of WhatsApp users were identified (taking into account the non-significance of the LMR LRT in the solution with five classes and the clinical meaning of the empirically derived classes, as shown in Table 4).

Class 1 (*n* = 77; 23%) had low scores for all the indicators (except for DERS). Class 2 (*n* = 37, 11%) spent an average of 2.68 h on WhatsApp but had the highest values for all other observed variables. The third class (*n* = 85, 25%) had the highest levels for time spent on WhatsApp but mild scores for all other indicators. Class 4 (*n* = 141, 41%) spent little time on WhatsApp but had mild scores for all other indicators (Table 7).

A statistically significant difference in distress scores (CORE-YP) was found between classes for CORE-YP (F(3, 336) = 8.222, *p* < 0.001; Class 2 > Classes 1, 3, and 4).

## 4. Discussion

Although several studies in the last decade have examined the characteristics of smartphone use, the link between mobile screen time and psychological characteristics among adolescents has remained elusive. The current study used LCA to explore homogeneous profiles of smartphone and social media use and to explore their association with mental health distress.

Regarding overall screen time, our LCA results suggested that a three-class model fit well to the data. Participants in the first class had the lowest scores for all the indicators (except for DERS); participants in the third class had the greatest scores on all indicators; and participants in the second class had social media use intensity scores similar to Class 3, whereas the other indicators were similar to those in Class 1. These results suggest that adolescents with heavy, objective, daily smartphone use (8 h—Class 3) showed greater psychological distress, greater problematic social media use, a tendency towards social comparison and moderate difficulties regulating their emotions. Nineteen percent of adolescents in our sample belonged to this group, whereas the majority of participants use smartphones for an average of six hours daily, reporting moderate social media activities (i.e., amount of active social media use) but no elevated dysfunctional characteristics.

Overall, the current study suggests that different classes of smartphone users may be identified through LCA, providing a comprehensive view of the patterns of adolescents’ smartphone use. Our findings are also consistent with those reported by previous empirical studies which explored the classes of smartphone usage among adolescents [69]. For example, a study with a sample of Swiss adolescents [70] identified different classes of social media users, of which the high social use class reported the greatest problematic smartphone use levels and lowest quality of life. Another study [71] examined mobile screen-based media use in a Chinese sample of adolescents and identified a high-risk group reporting low physical activity, high level of self-harm and media use. The present study adds to this literature by expanding on the role of psychological variables such as emotion dysregulation and online social comparison, which may represent relevant facets of problematic smartphone use. The group with a heavy, objective, daily smartphone use and greater psychological distress differs considerably from the other two groups in terms of their ability to experience and regulate their internal states. Our results seem in line with prior research suggesting that the use of dysfunctional emotion regulation strategies could have an impact on problematic smartphone use as a way of coping with negative emotions, recording greater scores for maladaptive CER strategies, such as rumination, self-blame, blaming others and catastrophizing, in the group of ‘problematic’ users [11]. Our results are also consistent with a recent study [39] which showed that maladaptive emotion regulation strategies can trigger problematic smartphone use and social media in socially anxious subjects. These findings are also consistent with the evidence of the role of emotion dysregulation in problematic technology use, suggesting that young people reporting dysfunctional strategies in emotional regulation and abnormal use of technology often present behavioral difficulties or higher rates of negative mood symptoms [45].

Consistent with the compensatory internet use theory [38], our results show that adolescents may use their smartphones extensively in order to escape from their unregulated, negative, affective states. Given the relevance of the emotion regulation function in the overall functioning of the individual and the adolescent, in particular, this finding merits further investigation and should be regarded as a relevant target for future intervention to lessen mental health problems. Moreover, our findings provide initial evidence that greater tendencies towards online social comparison may be relevant for adolescents belonging among the problematically connected users. To date, research showing that exposure to positive posts on social media can negatively elicit an individual’s emotional responses has relied mostly on adult samples [52,53] and further research is needed to examine the interplay between social comparison and addictive internet use among young people [16,55]. However, to identify adolescents who might be more at risk for excessive smartphone use, there is a need for renewed focus when it comes to tracing the effects of smartphone use in the long run [5]. The current study only provides cross-sectional data that prevent us from examining the negative impact of excessive mobile screen time in the long term.

In this study, we also adopted LCA to examine different groups of adolescents based on their time spent on specific apps, such as Instagram, TikTok and WhatsApp, and the findings were mixed.

Regarding Instagram use, a four-class model fitted well to the data. The problematic class (13% of the sample—Class 2) had the highest scores for psychological distress, emotion dysregulation, social media addiction and social comparison, and adolescents in this group spent an average of 2.77 h daily on the app. However, the heavy users (16%—Class 3) who spent 8 h a day on Instagram reported only mild to moderate scores for dysfunctional psychological variables. Class 1 (22%) had the lowest scores for all the indicators (except for DERS), and Class 4 (49%) spent an average of 1.98 h on Instagram but had mild scores for all other indicators.

Regarding TikTok, the results of a LCA three-class model showed that heavy users (20% of the sample) spent 9 h daily on the app but reported levels of addictive social media use, online social comparison and emotion dysregulation close to those of low users (60% of the sample—Class 2), who spent two hours a day on TikTok. The other class (Class 1—20% of the sample) had the lowest scores for all the indicators. Taken together, these results highlight that heavy users of social media apps which mainly focus on video and photo-sharing do not report high dysfunctional psychosocial characteristics, and that a risk of high social media use does not overlap fully with vulnerable psychological characteristics. These findings seem to be in line with those showing no direct relationship between Instagram use (research on TikTok is still in its infancy) and poor mental well-being [59,72]. Our findings seem to suggest that high social media intensity (i.e., amount of active social media activities) is associated weakly with the amount of screen time, consistent with previous findings which highlighted the need to differentiate between the purpose or motivations of screen use and the exposure time [34,73]. Moreover, it could be speculated that other variables may be relevant to describe the psychological characteristics of heavy users. Different motives for social media use, such as escapism, social communication, and body image feedback, were shown to be related to social media engagement [74]. It is also worth noting that adolescent users of Instagram or TikTok may build a physically attractive online self-presentation by sharing photos/videos and receiving feedback in the form of likes and comments [54]. This kind of ideal self-presentation may have positive rather than negative effects on an adolescent’s distress.

Finally, regarding WhatsApp, a four-class model fitted well to the data. Problematic WhatsApp users (11% of the sample—Class 2) had the highest scores for psychological distress, social comparison and social media addiction, but spent 2.68 h a day on the app. On the other hand, heavy WhatsApp users (25%—Class 3) spent 9.22 h a day on the app, but reported mild to moderate levels of distress, emotion dysregulation and social media addiction. Class 1 (23% of the sample) had low scores for all the indicators (except for DERS), and Class 4 (41% of the sample) spent little time on WhatsApp but had mild scores for all other indicators.

WhatsApp is an instant messaging service used as a tool to ‘stay in touch’ with others virtually. The amount of time connected to the app, which is very high in our group of ‘heavy users’, is probably not sufficient to understand the function that the app takes on in the adolescent’s behavior. This evidence further confirms the need to investigate what kind of WhatsApp interactions (e.g., single or group conversations), with whom (peers or others) and what experiences accompany such relational exchanges.

Overall, our results suggest that excessive mobile screen time can be a correlate of psychological dysfunction for a subgroup of adolescents, characterized by emotion dysregulation and a high tendency towards both online social comparison and social media addiction. However, objective social media use per se does not represent an indicator of an adolescent’s problematic psychological characteristics, as suggested by prior studies [26,75]. However, when we examine time spent on specific apps, it seems that heavy users are not necessarily the same as psychological problematic users; this pattern of results calls for research examining mobile screen time in more nuanced and diverse ways, distinguishing between the use of different apps. This is especially important when studying adolescents’ online behavior, given that many of them are now permanently online and mobile apps are part of their own “digital identity” [76]. To date, there are inconsistent findings on the potential interplay between problematic smartphone use and social media use [13] or the overlap between a maladaptive use of specific social media apps. Further research is needed to explore what social networking apps might trigger problematic smartphone use among adolescents and which psychological profile may be most at risk [21,77].

The strengths of the current study are the use of digital trace data to explore clusters of smartphone use, i.e., the number of calls, text messages, social media use, gaming, Internet use, as well as the focus on the use of specific apps, such as Instagram, TikTok and WhatsApp. Prior evidence is largely based on adolescent data focused on self-reported smartphone use, which has a risk of over- or under-estimation; smartphone use is mainly examined as a global phenomenon, irrespective of the apps used [4]. Despite these strengths, the current study has some noticeable limitations. Firstly, the findings are cross-sectional, and causal links among the study variables cannot be detected. Secondly, although the adoption of an objective measure of using apps is an asset in this study, which allowed us to avoid self-reported and biased estimates of smartphone use, there are further indicators of smartphone use, such as mobile screen unlocking [78], which may be used to assess adolescent mobile usage. Furthermore, we cannot exclude concerns that adolescents might not report the right number of hours in a day spent on screen activities from their phone, due to social desirability norms. Thirdly, in the current study, we did not measure smartphone addiction with a validated tool. Although excessive time spent on smartphones can be a marker of problematic mobile use [13], the adoption of a specific measure of smartphone addiction may allow one to assess different patterns of problematic mobile use such as craving, withdrawal or impaired daily functioning. Finally, we did not use external criteria (e.g., demographic variables, psychological constructs, behavioral outcomes) to validate the LCA results.

The current findings have relevant clinical implications for mental health practitioners in terms of developing prevention activities and interventions for problematic social media use and its co-occurrence. In particular, work aimed at limiting the risk factors associated with problematic social media use may focus on facilitating emotional regulation skills and promoting the critical use of social media.

## 5. Conclusions

There is a growing interest in understanding the role of smartphones and digital tools in adolescent development. In agreement with previous research [6,15,18,21,26], it is essential to move away from the definition of ‘smartphone addiction’ and dive deeper into which variables contribute to adolescents’ health-damaging use. The most recent scientific landscape aims to understand the co-occurrence of certain characteristics of adolescent functioning in order to allow the scientific, clinical and social intervention scene to better construct useful interventions to lessen problematic behaviors related to social media use. Moreover, longitudinal research on the link between social media use and mental distress reported mixed findings [23,79], and there is a call to examine intraindividual and interpersonal factors that might explain the characteristics of psychological distress during adolescence. In line with this evidence, our contribution emphasizes the involvement of emotional regulation and online social comparison in the definition of dysfunctional social media use profiles.

## Figures and Tables

**Table 1 ijerph-20-06439-t001:** Participants’ socio-demographic characteristics.

	Participants (*n* = 340)
Age, M (SD)	15.61 (1.19)
Gender, *n* (%) females	130 (38.3)
Engagement status, *n* (%)	
Single	255 (75.0)
In a relationship	85 (25.0)
City of residence, *n* (%)	
Naples	190 (55.9)
Palermo	150 (44.1)
School class, *n* (%)	
High school, I class	74 (21.8)
High school, II class	70 (20.6)
High school, III class	125 (36.8)
High school, IV class	71 (20.9)
Social media usage ^1^, *n* (%) yes	
Instagram	306 (90.0)
Facebook	38 (11.2)
TikTok	283 (83.2)
Snapchat	19 (5.6)
Twitter	55 (16.2)

^1^ Participants may select more than one option.

**Table 2 ijerph-20-06439-t002:** LCAs model fit indices (time on the smartphone, social media addiction, social media use intensity, online social comparison and emotion dysregulation as indicators).

Model	BIC	ssaBIC	Entropy	LMR LRT	BLRT	Smallest Class Size
#1	10,968.430	10,936.708	-	-	-	-
#2	10,815.171	10,764.416	0.888	−5455.070 ***	−5455.070 ***	26%
#3	10,736.599	10,666.811	0.820	−5360.954 ***	−5360.954 ***	19%
#4	10,622.118	10,640.507	0.821	−5304.181	−5304.181 ***	14%
#5	10,719.037	10,611.183	0.853	−5283.059	−5283.059 ***	5%

Note: LCA = Latent Class Analysis; BIC = Bayesian information criterion; ssaBIC = sample size adjusted BIC; LMR LRT = Lo–Mendell–Rubin likelihood ratio test; BLRT = bootstrap likelihood ratio test; # = number of classes; *** *p* < 0.001.

**Table 3 ijerph-20-06439-t003:** Descriptive analyses for the three LCA classes (time on the smartphone, emotion dysregulation, social media addiction, social media use intensity, and online social comparison as indicators).

	Time on Smartphone	DERS	BSMAS	Social Media Use Intensity	INCOM
Class 1	5.34 (0.26)	47.02 (1.42)	11.99 (0.42)	13.09 (0.56)	32.04 (0.92)
Class 2	6.19 (0.18)	45.84 (1.06)	13.52 (0.34)	25.08 (0.29)	33.63 (0.59)
Class 3	8.00 (0.26)	58.92 (1.98)	21.57 (0.71)	26.08 (0.38)	41.06 (1.09)

Note: Means and standard deviations are reported in the table; DERS = Difficulties in Emotion Regulation Scale; BSMAS = Bergen Social Media Addiction Scale; INCOM = IOWA-Netherlands Comparison Orientation Measure.

**Table 4 ijerph-20-06439-t004:** LCAs model fit indices (time spent on TikTok/Instagram/WhatsApp, social media addiction, social media use intensity, individual’s tendency to evaluate him/herself through comparison with others, and emotion dysregulation as indicators).

	Model	BIC	ssaBIC	Entropy	LMR LRT	BLRT	Smallest Class Size
TikTok	#1	11,117.991	11,124.558	-	-	-	
	#2	10,970.892	10,920.137	0.960	−5548.995 ***	−5548.995 ***	20%
	#3	10,903.366	10,833.577	0.887	−5438.814 **	−5438.814 ***	20%
	#4	10,867.070	10,778.249	0.921	−5387.564 **	−5387.564 ***	5%
	#5	10,839.477	10,731.326	0.855	−5351.930 **	−5351.93 ***	5%
Instagram	#1	11,036.741	11,005.019	-	-	-	
	#2	10,886.968	10,836.213	0.882	−5489.226 ***	−5489.226 ***	26%
	#3	10,797.626	10,727.838	0.877	−5396.852 ***	−5396.852 ***	17%
	#4	10,769.575	10,680.753	0.825	−5334.695 *	−5334.695 ***	13%
	#5	10,760.141	10,652.286	0.836	−5303.182 *	−5303.182 ***	4%
WhatsApp	#1	11,178.539	11,146.817	-	-	-	
	#2	11,031.376	10,980.621	0.880	−5560.125 ***	−5560.125 ***	26%
	#3	10,936.080	10,866.291	0.899	−5451.446 *	−5451.446 ***	23%
	#4	10,920.934	10,832.113	0.843	−5403.921 **	−5403.921 ***	11%
	#5	10,874.429	10,766.575	0.865	−5362.547	−5362.547 ***	6%

Note: LCA = Latent Class Analysis; BIC = Bayesian information criterion; ssaBIC = sample size adjusted BIC; LMR LRT = Lo–Mendell–Rubin likelihood ratio test; BLRT = bootstrap likelihood ratio test; # = number of classes; * *p* < 0.05; ** *p* < 0.01; *** *p* < 0.001.

**Table 5 ijerph-20-06439-t005:** Descriptive statistics (means and SE) for the three LCA classes (time spent on TikTok, emotion dysregulation, social media addiction, social media use intensity, and online social comparison as indicators).

	Time on TikTok	DERS	BSMAS	Social Media Use Intensity	INCOM
Class 1	1.84 (0.23)	45.77 (1.52)	11.46 (0.49)	12.86 (0.69)	31.95 (1.13)
Class 2	2.02 (0.11)	48.82 (0.97)	15.21 (0.37)	25.07 (0.30)	35.35 (0.57)
Class 3	9.01 (0.018)	51.40 (1.73)	16.58 (0.64)	23.50 (0.70)	35.51 (0.97)

Note: DERS = Difficulties in Emotion Regulation Scale; BSMAS = Bergen Social Media Addiction Scale; INCOM = IOWA-Netherlands Comparison Orientation Measure.

**Table 6 ijerph-20-06439-t006:** Descriptive statistics (means and SE) for the four LCA classes (time spent on Instagram, emotion dysregulation, social media addiction, social media use intensity and online social comparison as indicators).

	Time on Instagram	DERS	BSMAS	Social Media Use Intensity	INCOM
Class 1	1.64 (0.21)	47.27 (1.49)	11.81 (0.44)	12.87 (0.58)	31.94 (0.96)
Class 2	2.77 (0.58)	59.29 (2.21)	21.91 (1.21)	25.84 (0.51)	40.33 (1.45)
Class 3	8.00 (0.35)	49.13 (2.32)	16.17 (0.97)	24.69 (0.64)	36.34 (1.47)
Class 4	1.98 (0.14)	45.91 (1.36)	13.42 (0.42)	25.07 (0.33)	33.73 (0.69)

Note: DERS = Difficulties in Emotion Regulation Scale; BSMAS = Bergen Social Media Addiction Scale; INCOM = IOWA-Netherlands Comparison Orientation Measure.

**Table 7 ijerph-20-06439-t007:** Descriptive statistics (means and SE) for the four LCA classes (time spent on WhatsApp, emotion dysregulation, social media addiction, social media use intensity and online social comparison as indicators).

	Time on WhatsApp	DERS	BSMAS	Social Media Use Intensity	INCOM
Class 1	2.85 (0.29)	47.33 (1.47)	11.86 (0.45)	12.70 (0.55)	32.07 (1.00)
Class 2	2.68 (0.33)	59.52 (2.44)	21.68 (1.27)	25.59 (0.57)	40.73 (1.40)
Class 3	9.22 (0.13)	49.34 (1.51)	15.80 (0.59)	24.93 (0.44)	35.42 (0.91)
Class 4	2.61 (0.16)	45.73 (1.45)	13.51 (0.38)	25.03 (0.35)	33.85 (0.74)

Note: DERS = Difficulties in Emotion Regulation Scale; BSMAS = Bergen Social Media Addiction Scale; INCOM = IOWA-Netherlands Comparison Orientation Measure.

## Data Availability

Data are available from the corresponding author via email.

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
