# Peer review of "Time Spent on Mobile Apps Matters: A Latent Class Analysis of Patterns of Smartphone Use among Adolescents"

_ijerph, 2023, doi:10.3390/ijerph20156439_

Round 1

Reviewer 1 Report

Area of concern:

1. Please provide an academic basis (aside from the scores) from where the authors can determine the Classes used to classify the scores of the respondents. True, LCA was used. But, to determine leading labels such as "healthy users", "heavy users and problematic" etc. seems contrary to objectivity. I would suggest the authors change the labels to only include Class 1 and the characteristics, but to refrain from use of the labels. Thank you. 

Suggested areas for improvement:

1. The authors make a strong statement of overuse of social media in the latter part of the first introductory paragraph with no citation to support the statement. May I urge the authors to add relevant citations to support these statistics? Thank you. 

1. What is the main question addressed by the research?

The main question addressed by the authors is the impact of the use of social media on the rest of their health. I believe the authors have managed to find a niche in this area of research.
2. Do you consider the topic original or relevant in the field? Does it
address a specific gap in the field?

The topic is original in that it has found a niche. I would, however, caution that with the continued use of social media there will be a deluge of research on this topic. The authors, and other scholars in this field ought to do their utmost to keep abreast of these developments.
3. What does it add to the subject area compared with other published
material?

By comparing the types of social media the adolescents subject themselves to the authors have been able to give us a glimpse into the idea that it is not social media per se that should be of concern but the type of content that is within the social media. 
4. What specific improvements should the authors consider regarding the
methodology? What further controls should be considered?

Nothing to add on this point.
5. Are the conclusions consistent with the evidence and arguments presented
and do they address the main question posed?

The conclusions are consistent, but I would also advise that the authors be cognizant of the field of technology, social media and social media use expanding each day. The authors should be more cognizant of delivering conclusions that encompass this reality.
6. Are the references appropriate?

As far as I could tell, yes. 
7. Please include any additional comments on the tables and figures.

None. Thank you. 

1. There are often problematic statements that lead to confusion of meaning, for example, lines 104-107. May I please suggest a thorough proofreading of the manuscript. This will lend further clarity to the message/meaning of the authors in the manuscript. Thank you. 

Author Response

Reviewer 1

Comments and Suggestions for Authors

Area of concern:

1. Please provide an academic basis (aside from the scores) from where the authors can determine the Classes used to classify the scores of the respondents. True, LCA was used. But, to determine leading labels such as "healthy users", "heavy users and problematic" etc. seems contrary to objectivity. I would suggest the authors change the labels to only include Class 1 and the characteristics, but to refrain from use of the labels. Thank you.

Response: Thank you for your comment. Following your suggestion, in the revised manuscript we have changed the labels of the classes refraining from using labels and including only the number of the classes and their characteristics.

Suggested areas for improvement:

2. The authors make a strong statement of overuse of social media in the latter part of the first introductory paragraph with no citation to support the statement. May I urge the authors to add relevant citations to support these statistics? Thank you.

Response:  Thank you for your comment. The following sources of the presented data have been included:

  • Pew Research Center. Available online: https://www.pewresearch.org/internet/2018/05/31/teens-social-media-technology-2018/ (accessed on 02/03/2023).
  • Smartphones in Italy 2023. Available online: https://www.statista.com/study/41265/smartphones-in-italy/https: //www.statista.com/search/?q=smartphone+adolescent+2022&Search=&qKat=search&newSearch=true&p=1 (accessed on 02/0305/2023).
  • 17° Rapporto sulla comunicazione. I media dopo la pandemia.2021, Milano: Franco Angeli.

3. What is the main question addressed by the research?

The main question addressed by the authors is the impact of the use of social media on the rest of their health. I believe the authors have managed to find a niche in this area of research.

Response: Thank you for your comment.

4. Do you consider the topic original or relevant in the field? Does it address a specific gap in the field?

The topic is original in that it has found a niche. I would, however, caution that with the continued use of social media there will be a deluge of research on this topic. The authors, and other scholars in this field ought to do their utmost to keep abreast of these developments.

Response: Thank you for your comment.

5. What does it add to the subject area compared with other published material?

By comparing the types of social media the adolescents subject themselves to the authors have been able to give us a glimpse into the idea that it is not social media per se that should be of concern but the type of content that is within the social media.

Response: Thank you for your comment.

6. What specific improvements should the authors consider regarding the methodology? What further controls should be considered?

Nothing to add on this point.

Response: Thank you for your comment.

7. Are the conclusions consistent with the evidence and arguments presented and do they address the main question posed?

The conclusions are consistent, but I would also advise that the authors be cognizant of the field of technology, social media and social media use expanding each day. The authors should be more cognizant of delivering conclusions that encompass this reality.

Response: Thank you for your comment. The conclusions have been modified as follows:

“In agreement with previous research [6, 15, 18, 21, 26], it is essential to move away from the definition of 'smartphone addiction' and dive deeper into which variables contribute to adolescents' health-damaging use. In light of this, more robust studies of the changing roles of emerging technology in young people’s lives should consider that the field of technology, social media and social media use expanding each day and, thus, use data that afford a more detailed, accurate, and updated look into the individuals engaging with the technology.”

8. Are the references appropriate?

As far as I could tell, yes.

Response: Thank you for your comment.

9. Please include any additional comments on the tables and figures.

None. Thank you.

Response: Thank you for your comment.

Comments on the Quality of English Language

10. There are often problematic statements that lead to confusion of meaning, for example, lines 104-107. May I please suggest a thorough proofreading of the manuscript. This will lend further clarity to the message/meaning of the authors in the manuscript. Thank you.

Response: The original manuscript was proofread by a native English speaker. However, following your comment, we had a second proofreading of this revised manuscript, in order to improve its language clarity.

Reviewer 2 Report

This study entitled, “Time spent on mobile apps matters: A latent class analysis of patterns of smartphone use among adolescents,” is well-structured and carries out a latent class analysis that address an important research gap. The research design, instruments, and statistical analyses are well-explained and easy to understand.

However, I have some comments to improve this manuscript.

Introduction

·       Add a justification for why the authors selected only five social media apps for analysis. This can be based on the popularity of the apps, previous research on their effects, or other relevant factors.

Literature

·       Check the use of acronyms SU, SMU, and SU again. In some places, SU defines two different ways.

·       Formulate clear research questions or hypotheses using bullet points.

Methods

·       Provide more detail on the sampling technique used to recruit participants. For example, describe the inclusion and exclusion criteria, the source population, and the recruitment strategy.

·       Define the parameter estimation method used in the LCA. For example, if the authors used maximum likelihood estimation, they can explain how it works and why it was chosen.

·       Provide a range of values for the entropy measure that indicates good model fit. For example, the authors can refer to previous research or expert guidelines or recommendations to determine a meaningful threshold.

Results

·       Provide a more detailed description of how the LCA results were validated using external criteria. Specify the types of external criteria used (e.g., demographic variables, psychological constructs, behavioral outcomes), the statistical methods used to compare the identified classes (e.g., chi-square tests or regression or ANOVA), and the results of these analyses. Consider presenting the results in a separate table or figure.

Discussion

·       The authors could summarize the main findings of the study, including the identified latent classes, and their characteristics. However, please add more about their association with the external criteria. And discuss more the implications of these findings for theory, and practice.

By addressing the comments and suggestions, the study can be improved and potentially be considered acceptable.

Author Response

Reviewer 2

Comments and Suggestions for Authors

This study entitled, “Time spent on mobile apps matters: A latent class analysis of patterns of smartphone use among adolescents,” is well-structured and carries out a latent class analysis that address an important research gap. The research design, instruments, and statistical analyses are well-explained and easy to understand.

However, I have some comments to improve this manuscript.

Introduction

Add a justification for why the authors selected only five social media apps for analysis. This can be based on the popularity of the apps, previous research on their effects, or other relevant factors.

Response: Thank you for your suggestion. In the revised manuscript, the selection of Instagram, Tik Tok and WhatsApp according to their popularity among Italian adolescents is supported by previous research (i.e., Horvath, J., Mundinger, C., Schmitgen, M.M., Wolf, N.D., Sambataro, F., Hirjak, D. Structural and functional correlates of smart- phone addiction. Addict Behav. 2020, 105, 106334. https://doi.org/10.1016/j.addbeh.2020.106334).

Literature

Check the use of acronyms SU, SMU, and SU again. In some places, SU defines two different ways.

Response: We apologize for the oversight. Considering your comment, which is consistent with a point raised by the third reviewer, in the revised manuscript we avoid acronyms and use long forms for each variable.

Formulate clear research questions or hypotheses using bullet points.

Response: In the revised manuscript we provided clear research questions using bullet points for each aim of the study, as follows:

“[…] the aims of the present study are twofold: 1) to empirically determine homogeneous groups of adolescents based on their patterns of smartphone use (i.e., daily overall time spent and time spent on smartphone applications), social media use (i.e., intensity of social media use and problematic social media use), and psychosocial variables (i.e., emotion regulation and online social comparison). For the purposes of this study, we examined the most widely used social networking services or instant messaging apps among adolescents such as Instagram, Tik Tok and WhatsApp [14]; and 2) to explore whether the adolescent empirically-derived profiles are associated with risky psychological outcomes. Regarding the first goal, we pose the following research questions:

  • what psychological clusters can be identified among adolescents based on their smartphone and social media activities?
  • do the participants’ psychological characteristics vary across the clusters, according to the types of smartphone application?

Regarding the second aim we pose the following research question:

  • do adolescents with more problematic patterns of smartphone use report higher level of psychological distress than adolescents with less problematic smartphone use?

Methods

Provide more detail on the sampling technique used to recruit participants. For example, describe the inclusion and exclusion criteria, the source population, and the recruitment strategy.

Response: Thank you for your comment. In the revised manuscript, we provided more information about the recruitment of the participants, as follows:

Three hundred and forty adolescents aged 13 to 19 (M = 15.61, SD = 1.19) were included in the study. Participants were recruited from two secondary schools (from the first to the fourth year of high school) in southern Italy (i.e., Palermo and Naples) from March to April 2022. Inclusion criteria were: (a) owning a smartphone, (b) parents having signed the study informed consent document, and (c) Italian language skills sufficient to understand the study questionnaires.

[…]. Information about the study and consent forms were sent to the teachers and the parents of all adolescents. It was specified that participation was voluntary, that the questionnaires were anonymous (i.e., data were anonymized with a single research identification number for each participant), that no pressure would be applied should they choose to return the questionnaire unfilled or incomplete and that they could withdraw from the study at any time. Participants received no compensation. Using an online survey, questionnaires were completed in the adolescents’ school classrooms, during regular school hours, under the supervision of research assistants, who supervised data collection and answered student questions.”

Define the parameter estimation method used in the LCA. For example, if the authors used maximum likelihood estimation, they can explain how it works and why it was chosen.

Response: Thank you for your comment. We used maximum likelihood estimation. This point has been clarified in the Plan of Data Analysis section, as follows:

For the first aim of the study, a first LCA was conducted in Mplus (version 7) using maximum likelihood estimation to classify the participants into homogeneous groups of smartphone and social media use patterns (i.e., time spent on the smartphone, social media addiction, social media use intensity, online social comparison, and emotion dysregulation). […] Maximum likelihood estimation is the process of estimating model parameters such that the resultant model generates values that are most likely to represent actual observed values [67].

Provide a range of values for the entropy measure that indicates good model fit. For example, the authors can refer to previous research or expert guidelines or recommendations to determine a meaningful threshold.

Response: A range of values for the entropy measure that indicates good model fit is now reported in the Plan of Data Analysis section, as follows:

The most suitable model had the following fit indices: BIC and ssaBIC should be lower; entropy should be larger (more specifically, entropy can range from 0 to 1 and values of .80 or greater provide supporting evidence that profile classification of individuals in the model occurs with minimal uncertainty [68]), the smallest class size should be > 5%; and LMR and BLRT should be significant [69].”

Results

Provide a more detailed description of how the LCA results were validated using external criteria. Specify the types of external criteria used (e.g., demographic variables, psychological constructs, behavioral outcomes), the statistical methods used to compare the identified classes (e.g., chi-square tests or regression or ANOVA), and the results of these analyses. Consider presenting the results in a separate table or figure.

Response: We did not collect data about external criteria in order to validate LCA results. As usually done in the literature, the best-fitting solutions have been selected by exploring LCA results in relation to statistical indexes (i.e., AIC, BIC, ssaBIC, the size of the smallest class size, entropy, and LMR-LRT), theoretical coherence (i.e., substantive interpretability of the trajectory classes and identification of trajectories without overfitting), and explanatory relevance. Nevertheless, the absence of external criteria has been reported as a limitation of the study, as follows:

Finally, we did not use external criteria (e.g., demographic variables, psychological constructs, behavioral outcomes) in order to validate LCA results”.

Discussion

The authors could summarize the main findings of the study, including the identified latent classes, and their characteristics. However, please add more about their association with the external criteria. And discuss more the implications of these findings for theory, and practice.

Response: Regarding the first point, the main findings of the study (i.e., identified latent class and their characteristics) are now summarized in the Discussion section, as follows:

Regarding the overall screen time, our LCA results suggested that a three-class model had the best fit to the data. Participants in the first class had the lowest scores on all the indicators (except for DERS); participants in the third class had the highest scores on all indicators; and participants in the second class had social media use intensity scores similar to the Class 3, whereas the other indicators were similar to those in Class 1.”

Moreover, the main findings on each application (i.e., Tik Tok, Instagram and WhatsApp) were described later in the Discussion section (i.e., lines 442-479), as follows:

Regarding Instagram use, a four-class model had the best fit to the data. The problematic class (13% of the sample – Class 2) had the highest scores on psychological distress, emotion dysregulation, social media addiction and social comparison, and adolescents in this group spent an average 2.77 hours daily on the app. However, the heavy users (16% - Class 3) who spent 8 hours a day on Instagram reported only mild to moderate scores on dysfunctional psychological variables. Class 1 (22%) had the lowest scores on all the indicators (except for DERS), and Class 4 (49%) spent an average 1.98 hours on Instagram but had mild scores on all other indicators.

Regarding TikTok, results of a LCA three-class model showed that heavy users (20% of the sample) spent 9 hours daily on the app but reported levels of social media addiction, online social comparison and emotion dysregulation close to those of low users (60% of the sample – Class 2), who spent two hours a day on TikTok. The other class (Class 1 – 20% of the sample) had the lowest scores on all the indicators. […] Finally, regarding WhatsApp, a four-class model fitted well to the data.  Problematic WhatsApp users (11% of the sample – Class 2) had the highest scores on psychological distress, social comparison and social media addiction but spent 2.68 hours a day on the app. On the other hand, heavy WhatsApp users (25% - Class 3) spent 9.22 hours a day on the app, but reported mild to moderate levels of distress, emotion dysregulation and social media addiction. Class 1 (23% of the sample) had low scores on all the indicators (except for DERS), and Class 4 (41% of the sample) spent little time on WhatsApp but had mild scores on all other indicators.

Regarding the second point, we did not use external criteria (see our response to your previous comment about this point).

Regarding the third point, the implications of the study are now summarized in the Discussion section, as follows:

“The present study’s findings have important practical implications for psychological and mental healthcare practitioners in developing prevention and intervention programs for problematic social media use, and its co-occurrence. In particular, work aimed at limiting the risk factors associated with problematic social media use may focus on facilitating emotional regulation skills and promoting the critical use of social media.”

By addressing the comments and suggestions, the study can be improved and potentially be considered acceptable.

Response: Thank you for your comments. Answering them allowed us to improve the manuscript and we hope you will find our answers satisfactory.

Reviewer 3 Report

Hello,

Thank you for the opportunity to review this paper. Overall, the paper was fairly easy to read, but I have a few larger and minor suggestions. First, the authors state a strength of this paper is the focus on specific applications (Instagram, tik tok, and WhatsApp). Discussing these applications more in-depth and relating them to the variables explored would be beneficial. In section 1.3, the authors mention that tik tok and Instagram might be used for social comparison, but What about WhatsApp or emotion regulation? Doing a better job connecting the application to the variables explored would show a much stronger understanding of the literature. Second, the authors first aim is to empirically determine homogeneous classes of adolescents based on patterns of social media use (SU), but to what end? I need clarification on why this is important and how determining these homogenous groups can help users reduce social distress. I also do not think the differences in distress scores are well presented. Below are some minor comments.

Introduction

- Growth in the smartphone market has also emerged in European countries. In Italy, 93.5% of adolescents aged 15 to 17 years access the Internet on a daily basis [3]. The second sentence does not support the growth of smartphone usage. In some schools, children access the Internet on a daily basis for class.

-Lines 36-38 could be rewritten for clarity. You could state the percent increase from 2016 to the present.

-If the article is below word count, I would suggest consistently using the long form of each short form. It would improve the readability of the article.

-Lines 86-97. Screen time can include time in front of a TV, gaming console, computer, tablet or cell phone. It may have nothing to do with social media. Ensuring accuracy or operational definitions in the terms used is essential in understanding this piece. You go on to make this distinction. This should be done earlier, and you should follow the terms in the paper

-You need to be careful using a term like well-being. There is a large body of literature on this concept, and it would be best if you stuck to the actual outcomes examined.

-After reading the introduction, I am unsure why this research is important. Can you please provide 1-3 sentences on what can be done with the results of this study?

Methods

-Was the questionnaire for demographics based on any other tools? For example, gender can be considered a social construct as it presents more on a spectrum were m/f the only option? Were these the only social media options?

-2.2.1 Is there any concern that students would not report the actual number from their phone? The authors claim this as a strength, but is there any evidence that the teen would report the actual number from their phone? I know I feel embarrassed when I see my average screen time. 

Discussion

-Line 374 is confusing, and why is compensatory capitalized?

-Is time spent on an app related to the intensity of the use? I think this should be mentioned. You treat all time spent on the app equally, but it is not. For example, I use Twitter on a daily basis, but rarely tweet and use it more to read news and to pass the time. This type of usage must be different for the people who are after likes, follows, or re-tweets.

-However, is used twice in three lines, 427-430. This weakens writing.

Minor edits required 

Author Response

Reviewer 3

Comments and Suggestions for Authors

Hello,

Thank you for the opportunity to review this paper. Overall, the paper was fairly easy to read, but I have a few larger and minor suggestions. First, the authors state a strength of this paper is the focus on specific applications (Instagram, tik tok, and WhatsApp). Discussing these applications more in-depth and relating them to the variables explored would be beneficial. In section 1.3, the authors mention that tik tok and Instagram might be used for social comparison, but What about WhatsApp or emotion regulation? Doing a better job connecting the application to the variables explored would show a much stronger understanding of the literature.

Response: Following your suggestion, we provided a more in-depth description of the relations between the examined mobile applications and psychosocial variables among adolescents. For example, we added the following to the Introduction:

Although some empirical evidence suggested that emotion dysregulation may be associated with problematic Facebook use [47-49], there is still a dearth of research on the association between emotion dysregulation and time spent on specific applications such as Instagram, TikTok and WhatsApp among adolescents”.

Regarding instant messaging apps, in the Introduction we expanded a bit on recent research which showed that problematic smartphone use and problematic WhatsApp use are strongly intertwined and tended to form a cluster of their own [Rozgonjuk et al., 2020], with a call for further studies on the differences between problematic smartphone use and a problematic use of specific apps which is still in its initial phase.

Second, the authors first aim is to empirically determine homogeneous classes of adolescents based on patterns of social media use (SU), but to what end? I need clarification on why this is important and how determining these homogenous groups can help users reduce social distress. I also do not think the differences in distress scores are well presented. Below are some minor comments.

Response: Thank you for your suggestion. In the revised manuscript, we attempted to make the rationale of the present study clearer, by stressing how it can fill some gaps in the literature. Regarding the link between social media use and psychological distress, the current study explored whether adolescents with more problematic pattern of smartphone use report higher level of psychological distress than adolescents with a less problematic smartphone use. We think that our results, although preliminary and based on cross-sectional data, can be helpful to identify some subgroups more at risk of both problematic smartphone use and psychosocial distress.

Introduction

- Growth in the smartphone market has also emerged in European countries. In Italy, 93.5% of adolescents aged 15 to 17 years access the Internet on a daily basis [3]. The second sentence does not support the growth of smartphone usage. In some schools, children access the Internet on a daily basis for class.

Response: Thank you for your comment. We clarified the second sentence, as following:

“Growth in smartphone market has also taken place in European countries. In Italy, more than 95% of adolescents aged 14 to 17 years own a smartphone and access the Internet on a daily basis [2].”

- Lines 36-38 could be rewritten for clarity. You could state the percent increase from 2016 to the present.

Response: The sentence has been removed from the manuscript.

- If the article is below word count, I would suggest consistently using the long form of each short form. It would improve the readability of the article.

Response: Following your suggestion, in the revised manuscript we avoid acronyms and consistently use only long forms.

- Lines 86-97. Screen time can include time in front of a TV, gaming console, computer, tablet or cell phone. It may have nothing to do with social media. Ensuring accuracy or operational definitions in the terms used is essential in understanding this piece. You go on to make this distinction. This should be done earlier, and you should follow the terms in the paper

Response: It is an important point. Following your suggestion, we provided in the Introduction clearer description of the term screen time as follows:

“Screen time is an umbrella term which includes a variety of devices (e.g., computer, TV, tablet, smartphone) and uses (e.g., social communication, gaming) and there is an ongoing debate about its usefulness in studying adolescent’s mental health [21,22].

 Moreover, we made an accurate definition of the terms used, both in the Introduction and Methods section, which were consistently adopted across the manuscript, as follows:

“In the present study we will examine objective trace data on both a) overall daily time spent on the smartphone for any activity, and b) daily time spent on Instagram, TikTok, and WhatsApp”.

- You need to be careful using a term like well-being. There is a large body of literature on this concept, and it would be best if you stuck to the actual outcomes examined.

Response: Thank you for your comment. In the revised manuscript, we have consistently referred to psychological distress, given that in the current study we examined this variable through the Young Person's CORE-OM (YP-CORE).

-After reading the introduction, I am unsure why this research is important. Can you please provide 1-3 sentences on what can be done with the results of this study?

Response: Following your suggestion, we attempted to clarify why this study can be relevant to the literature. In the section 1.4 we added the following:

“Specifically, to garner a better understanding of how smartphone and social media use relates to adolescent phenomena such as psychological distress, we have build upon previous research which used clustering techniques to detect specific profiles of smartphone or social media users [15,58,59], to explore whether distinct and meaningful user profiles can be identified”.

Methods

- Was the questionnaire for demographics based on any other tools? For example, gender can be considered a social construct as it presents more on a spectrum were m/f the only option? Were these the only social media options?

Response: Thank you for your comment. Regarding gender, the third alternative was “non-binary”. However, none of the participants selected this option and we did not report this in Table 1. Yes, the only social media options are those reported in Table 1. Variables related to social media use are: apps used, reported objective time of use of the smartphone and individual apps, and intensity of social media use.

- 2.2.1 Is there any concern that students would not report the actual number from their phone? The authors claim this as a strength, but is there any evidence that the teen would report the actual number from their phone? I know I feel embarrassed when I see my average screen time.

Response: Good point. In the limitation section we added that:

“Furthermore, we cannot exclude concerns that adolescents might not report the actual number of hours in a day spent on screen activities from their phone, due to social desirability norms”.

Discussion

-Line 374 is confusing, and why is compensatory capitalized?

Response: The term “compensatory” has been changed to lowercase throughout the manuscript. Moreover, we had a second proofreading of this revised manuscript, in order to improve its language clarity.

- Is time spent on an app related to the intensity of the use? I think this should be mentioned. You treat all time spent on the app equally, but it is not. For example, I use Twitter on a daily basis, but rarely tweet and use it more to read news and to pass the time. This type of usage must be different for the people who are after likes, follows, or re-tweets.

Response: Thank you for your suggestion. We highlighted the difference between the app usage time and usage intensity throughout the manuscript. In the discussion section, we added the following:

“Our findings seem to suggest that high social media use intensity (i.e., amount of active social media activities) is associated weakly with the amount of screen time, consistent with previous findings which highlighted the need to differentiate between purpose or motivations of screen use and the exposure time [34;74].

- However, is used twice in three lines, 427-430. This weakens writing.

Response: We had a second proofreading of this revised manuscript, in order to improve its language clarity.

Comments on the Quality of English Language

Minor edits required

Response: We had a second proofreading of this revised manuscript, in order to improve its language clarity.

Round 2

Reviewer 2 Report

I am now content with the authors’ revisions made to this manuscript in response to the necessary feedback. Congratulations. It is now ready for publication.